# Effect of an Acute Resistance Training Bout and Long-Term Resistance Training Program on Arterial Stiffness: A Systematic Review and Meta-Analysis

**DOI:** 10.3390/jcm10163492

**Published:** 2021-08-07

**Authors:** Roman Jurik, Aleksandra Żebrowska, Petr Stastny

**Affiliations:** 1Department of Sport Games, Faculty of Physical Education and Sport, Charles University in Prague, 162 52 Prague, Czech Republic; 2Institute of Sport Sciences, Jerzy Kukuczka Academy of Physical Education, 40-065 Katowice, Poland; a.zebrowska@awf.katowice.pl

**Keywords:** resistance training, hypertension, arterial pressure, arterial stiffness, pulse wave velocity

## Abstract

Resistance training (RT) and exercise is useful for preventing cardiovascular disease, systolic hypertension and stroke, which are associated with the stiffening of the larger central arterial system. The aim of this systematic review was to (a) understand the changes in arterial stiffness (AS) in various parts of the body measurement after acute RT bout and long-term RT, and (b) to determine the impact of exercise intensity on these changes in healthy individuals. A systematic computerized search was performed according to the PRISMA in PubMed, Scopus and Google Scholar with final selection of 23 studies. An acute RT bout led to a temporary increase in pulse wave velocity (PWV) regardless of the measurement method or intensity. A long-term RT at above an 80% repetition maximum (RM) have an ambiguous effect on PWV. A low-intensity RT or whole-body vibration training program decreased carotid–femoral PWV and brachial–ankle PWV (*d* = 1.02) to between 0.7 ± 1.4 ms^−1^ (*p* < 0.05) and 1.3 ± 1.07 ms^−1^ (*p* < 0.05) and improved other cardiac functions. A long-term RT of moderate (60–80% 1RM) or low intensity (<60% one-repetition maximum (1RM)) can decrease AS. Low and moderate intensity RT is beneficial to reduce high AS to prevent cardiovascular diseases.

## 1. Introduction

Arterial stiffness (AS) increases greatly with age, independently of atherosclerosis, hypertension or other diseases [1,2]. It happens as a result of the mechanical wear of the arterial wall, in which connective tissue changes. Increased AS represented by pulse wave velocity (PWV) is usually associated with cardiovascular disease (CVD) and might result in death [3,4]. At the same time, it appears to be a precursor of hypertension [5], with aging having a greater effect on central, rather than peripheral, arterial hemodynamics [6,7]. Strategies designed to prevent or retard the process of central arterial stiffening are, therefore, likely to hold substantial benefit in reducing the burden of CVD. Lifestyle changes are generally recommended for the nonpharmacological treatment of any form of CVD [7], especially a long-term resistance training (RT) program if it does not worsen AS in individuals who have, or are at risk for, CVD [8].

Aerobic and resistance training are both well researched and recommended by scientific studies because they influence cardiovascular function and blood pressure [9,10,11,12,13,14,15]. Moreover, it has been proven that aerobics, combined and isometric RT, improved PWV; furthermore, participants with greater arterial stiffness at baseline [16] showed enhanced AS improvement after high-intensity aerobic exercise (AE). Dynamic resistance training, however, had no significant effect [17]. The issue of exercise intensity has been evaluated in recent reviews of acute and long-term RT [18,19,20], where each review applied a different approach to separate exercise intensity thresholds. In one case, moderate intensity was defined as 50–69% of a one-repetition maximum (1RM) [18]; in another it was below 75% of 1RM [19]; and one defined moderate intensity as approximately 80% of 1RM high-intensity values [20]. This inconsistency causes confusion since general guidelines for preventive exercise in healthy (not resistance-trained) population suggest starting an RT at a low intensity (40–60% 1RM) and gradually moving to a moderate intensity (60–80% 1RM) [21,22].

The inconsistency in defining exercise intensity might be the reason for the conflicting conclusions about high-intensity RT in the literature: it might have a negative effect on AS [23,24,25,26]; it is associated with increased AS in young subjects [19]; it neither improves nor impairs AS in healthy subjects [20]; and it only impairs cardiovascular health in chronic interventions involving upper-body muscles [18]. Moreover, PWV values and changes depend on the measurement method, such as the body part selected, even though the carotid–femoral PWV is considered to be the gold standard value [27]. However, these reviews did not consider that exercise intensity in combination with the method of PWV measurement might have influenced the results.

Since the literature shows discrepancies both in methodology and attitude toward RT as a means to improve AS, the purpose of this study was (a) to understand the changes in AS at various parts of the body as measured by an acute RT bout and a long-term RT, and (b) to determine the impact of exercise intensity on these changes in healthy individuals.

## 2. Methods

This systematic review was conducted in accordance with the recommendations presented in the Preferred Reporting Items for Systematic Reviews and Meta-Analyses (PRISMA) statement [28]. The protocol for this systematic review was published in PROSPERO under registration number CRD42020194200.

### 2.1. Search Strategy

A systematic computerized literature search was performed using PubMed and Scopus and included studies published in English from 1966 until December 2020. The search criteria included the following terms: blood pressure AND hypertension OR cardiovascular disease OR hypotension AND resistance training OR strength training OR weightlifting OR bodybuilding OR exercise. The search was limited by article type, species, subject, language, age and text availability. A manual search was performed using the reviews, the reference lists of the selected articles and Google Scholar.

### 2.2. Types of Studies

The review considered cohort studies, analytical cross-sectional studies, randomized control trials, nonrandomized control trials, intervention studies, case-control studies and others that included AS and PWV as well as data on acute RT bout and long-term RT in adult healthy population. The review studies were used for manual searches of their reference list. Dissertations, theses, conference proceedings, conference monographies and other reviews were not included. Retrospective studies were not included because the area of interest requires performing experiments. The qualitative component also considered the type of acute RT bout and long-term RT and methodological designs. The reviewers, R.J. and P.S., screened all titles and abstracts after removing duplicates and retrieved and assessed the full texts of the eligible articles. Any discrepancies were resolved by discussion.

### 2.3. Types of Outcomes

The review considered studies that included the following outcome measures: the carotid–femoral PWV before and after an acute RT bout and long-term RT, femoral-ankle PWV before and after acute RT bout and long-term RT, brachial-ankle PWV before and after specific RT interventions and Beta stiffness index after an acute RT bout. The exclusion criteria were as follows: the full text was not available in English, the study did not contain an appropriate description of the measurement devices and procedures, the study did not include a proper exercise or AS task, and the study did not report how the raw data were processed.

### 2.4. Data Extraction and Evaluation

Data extraction included aspects of the study population, such as age and sex, specific aspects of the RT intervention (sample size, type of exercise performed, presence of supervision, frequency, and duration of each session), RT combination, outcome measurements and results, and the PWV values (Appendix A); however, the studies were not rejected if any part was missing. The PEDro (Physiotherapy Evidence Database) scale [29] was used to assess the methodological quality of a study based on general criteria, such as concealed allocation, intention-to-treat analysis, and adequacy of follow-up. These characteristics make the PEDro scale a useful tool for assessing the quality of the methodology [30]. Extraction was performed by two reviewers (R.J., P.S.). The lack of clarity during the extraction was resolved by discussion between them. The PEDro scale, based on a Delphi list [31], was used for all articles even if the trials had already been rated by trained evaluators of the PEDro database (http://www.pedro.fhs.usyd.edu.au/, December 2020. Methodological quality criteria are described in detail in Appendix A.

### 2.5. Statistical Analysis and Calculation of Effect Size

The treatment effects across the studies was calculated using weighted means and by the Cohen’s *d* effect size (ES) according to Wilson and Lipsey 2001 [32]. The mean difference and its standard deviation were extracted from original studies. The ES was calculated separately for each AS outcome as a pre- and post-intervention difference (d_post_) between treatment groups and controls (d_cont_). The effect sizes were calculated using the adjustment for sample size in each study and with calculation of a 95% confidence interval. All calculations were made by Practical Meta-Analysis Effect Size Calculator (D. Wilson, George Mason University, Fairfax VA, USA) at George Mason University. Wilson suggested that *d* = 0.2–0.49 be considered a small; 0.5–0.80, a medium; and above 0.8 a large effect size. An adjustment for a small sample bias was applied to each effect size.

## 3. Results

### 3.1. Study Characteristics

A systematic literature search identified 17,375 records; however, after duplicates were removed, 11,530 records were screened based on the title and abstract. The title and abstract screening resulted in 79 records that were eligible for full-text review. From these records, 22 studies satisfied the quality and exclusion criteria and were selected after full-text screening. Of these studies, 10 compared changes in AS and PWV as a result of an acute RT bout, and 12 studies compared changes in AS and PWV after a long-term RT, of which 7 studies compared carotid–femoral, 3 compared femoral-ankle and 5 compared brachial PWV (Figure 1).

### 3.2. Changes in Arterial Stiffness and Pulse Wave Velocity as a Result of an Acute Resistance Training Bout

The immediate reaction to an acute RT bout, regardless of intensity, is an increase in the PWV, as demonstrated by all studies [33,34,35,36,37,38,39,40,41,42] (Table 1). This conclusion comprised healthy men and women without any medication. The results showed that not only did classic whole-body training [33,35,38,39] increase PWV but so did eccentric RT [34] and traditional split upper/lower body or separate body parts [36]. Sixty minutes after an acute RT bout, values returned to baseline in four studies [33,35,36,39], but in a study conducted by Barnes [34], there was a significant increase in PWV in the carotid-femoral direction for 48 h.

An acute RT bout increased PWV and aortic AS equally in men and women, and Kingsley et al. (2017) [38] suggested that it may have a similar transient effect on the vasculature regardless of sex. However, it is important to note that while these responses may appear to have a negative effect on the cardio vasculature, it only provided information 10 min after acute RT.

Based on the results of Nitzsche et al. [39], an acute RT bout with moderate intensity (70% 1RM), low repetition number and long set pause led to the lowest deflections of the measured parameters compared with low (30% 1RM) and low-moderate (50% 1RM) intensity. These intensities significantly increased PWV immediately after acute RT. However, 75% 1RM should be discouraged to avoid acute increases in blood pressure and AS [41]. Acute RT using free-weights and weight machines produced similar increases in some measures of pulse wave reflection and aortic stiffness in resistance-trained individuals. Therefore, these data suggest that free weights or a weight machine, increases myocardial workload and aortic stiffness, and decreases myocardial perfusion for at least 10–20 min after RT is completed [42].

An acute RT bout increased AS, blood pressure, and heart rate only in the short term, however, as expected, most of the RT values returned to baseline for healthy men and women within 60 min (Table 1).

### 3.3. Changes in Arterial Stiffness and Pulse Wave Velocity as a Result of a Long-Term RT

High-intensity RT above 80% 1RM showed no positive effect on carotid–femoral PWV in Cortez-Cooper et al. [43] or Croymans et al. [44] (*d* = −0.0822, with control groups *d* = 0.9643) (Table 2, Figure 2 and Figure 3). Croymans et al. [44] found no significant changes after a high-intensity 12-week RT and Au et al. [45] discovered that central AS was reduced after RT with no effect on local carotid artery distensibility or left ventricular mass, regardless of the load lifted during RT. The study by Cortez-Cooper et al. showed that RT increased carotid-femoral, but decreased femoral-ankle, PWV [43] (Figure 3). Therefore, RT had an ambiguous effect on PWV.

Results in studies of moderate intensity (60–80% 1RM) showed a PWV decrease with large effect sizes for carotid–femoral (*d* = 1.0624), femoral–ankle (*d* = 1.3662) but an increase in brachial–ankle (*d* = 1.4094) [45,46,47,48,49,50] (Figure 2, Figure 3, Figure 4 and Figure 5). All studies reporting carotid–femoral PWV showed a positive effect from RT training [49,52,53]. Brachial-ankle PWV increased significantly from baseline (*p* < 0.05) in the upper limb [47] or showed a small improvement in the lower-limb training group [48]. A moderate-intensity RT consisting of progressively increased intensity without a concurrent rise in training volume did not increase central or peripheral AS in the study by Casey et al. [46].

When a long-term RT was applied at low intensity, AS and PWV improved in five studies [45,51,52,53,54], and there was no report of a PWV increase in any types of measurement. The rest of the studies evaluated the effect of low intensity whole-body vibration training on special platforms [52,53]. Figueroa et al. [52] reported improvements in AS in his studies, which were subsequently confirmed by Lai et al. [53]. When pre and post differences between treatment and control groups were compared, we got values with high effect sizes for femoral–ankle PWV (*d* = 1.3662) and brachial–ankle (*d* = 1.0162) [45,51,52,53,54]. From those results, it can be clearly stated that a low-intensity RT (<60% 1RM) reduced AS in all cases regardless of femoral–ankle or carotid–femoral measurement (Figure 2, Figure 3, Figure 4 and Figure 5).

## 4. Discussion

Because physical activity protects against CVD [12,55,56,57], AE has been suggested as a nonpharmacological treatment to improve cardiovascular function in both younger [58,59] and older individuals [60,61]. The results of this study showed that the same general recommendation is relevant for long-term RT; therefore, it should be recommended as an important part of a complex exercise program, which is what the American College of Sports Medicine has done [62], especially concerning RT in combination with aerobic and whole-body vibration interventions. Long-term RT have favorable effects on muscular power and strength, blood pressure, and muscle function [63] as well as for detailed health parameters like AS [64]. Our results are in agreement with recent a systematic review, which reported large (*d* = −1.49 to −1.20) and moderate (*d* = −1.07) decreases in AS after a long-term RT [18]. On the other hand, there is still ambiguous evidence for high-intensity RT. The acute RT bout should be applied with caution because it temporarily increases PWV regardless of the measurement method and intensity. This has to be considered especially when a RT is applied nonperiodically.

Changes in AS were evoked after the end of an acute RT bout, whereas this study expected that those changes would return within an hour after training [33,35,36,39,41]. Deviation from basic AS values were expected in all training sessions. These findings are of clinical importance because they showed that the acute effect of exercise on AS was probably functional rather than structural [65]. The mechanism was probably decreased peripheral resistance and vascular muscle tone along with vasodilation, which increased arterial compliance in the muscular arteries, and increased blood pressure and cardiac output [66]. Barnes et al. [34] included acute eccentric exercise, which remains unexplored in this regard. A healthy man participated in a bilateral leg-press eccentric RT and unilateral elbow flexor eccentric RT. The exercises were performed at 110% of each subject’s one-repetition maximum. Moreover, exercises that overcame maximum resistance raised blood pressure to dangerous values in healthy individuals for 48 h after acute RT; therefore, high intensity RT cannot be considered appropriate for preventing increased AS, which is same recommendation for patients with cardiovascular disease [67,68,69].

Okamoto, Masuhara and Ikuta [70] found no significant change in AS after eight weeks of RT, which consisted of biceps curls. The load was set to 100% of the 1RM for eccentric muscle contraction and 80% of the 1RM for concentric muscle contraction. This study did not confirm the negative effect of eccentric or concentric RT with maximal and submaximal resistance on AS in healthy females. In these cases, further studies are needed to address the effects of eccentric and concentric RT on AS in one training session and in a long-term program, especially of moderate or low intensity. The indisputable advantage of a long-term RT compared to an acute RT bout is the influence on body composition, blood pressure, heart rate and other health parameters [65,71,72]. On the other hand, an acute RT bout can be used to determine how a patient responds to selected training parameters and whether there is a negative reaction to the program, such as a dangerous increase in blood pressure and AS or extreme muscle pain.

Saz-Lara et al. [65] suggested that exercise is effective in reducing AS in the long term, but the results of this review were not uniform. This might be because practical recommendations of the results are dependent on exercise intensity, and alas, the method of PWV measurements. Moderate-intensity RT uses resistances between 60% and 80% of 1RM, the most common, and decreased AS in studies with carotid-femoral PWV methods [46]. However, that brachial-ankle PWV after RT did not increase after exercising lower limbs, but was significantly increased in the upper limb RT [47]. The RT program in the upper limb group consisted of chest presses, biceps curls, seated rows, shoulder presses, and lat pull downs in consequence order. Previously, upper-body exercise has been reported to produce greater increases in arterial pressure, heart rate and total peripheral resistance than lower-body exercise [73]. Therefore, it is possible to hypothesize that the differences in baPWV to upper limb RT and lower limb RT are based, on the degree of activation of the sympathetic nervous system. A number of studies have provided evidence that increased sympathetic nervous system tone plays a crucial role in constricting dilated skeletal muscle blood vessels [73]. On the other hand, a very similar situation probably occurred in another study by Okamoto et al. [48], where moderate intensity did not improve brachial–ankle PWV in the RT group with slow lifting and quick lowering training [48]. However, if we included lower body exercises or full-body training, for example in the form of circuit training, we could observe an improvement in AS [54]. For future research, it is necessary to focus on the exercise selection and order [73,74,75]. It is well known that moderate intensity (60–80% 1RM) RT is characterized by muscular hypertrophy [76,77], which was not limited following progressive adaptation in healthy individuals. When we reduced resistance to 40–60% 1RM, we observed a significant improvement in AS and PWV [45,51,54,78,79]. Low-intensity RT (<60% 1RM) in various forms and parameters generally led to a decrease in brachial–ankle, femoral–ankle and carotid–femoral PWVs. Low-intensity RT with short inter-set rest (30 s) and 10 repetitions during five sets reduced the baPWV (from 1093 ± 148 to 1020 ± 128 cm/s, *p* < 0.05) and improved vascular endothelial function (brachial flow-mediated dilation) [51]. Training on vibration platforms is a safe alternative, which also has a positive effect on blood pressure, body composition and strength parameters. Due to its versatile use, it can be used by adults and seniors with prehypertension or hypertension [80,81,82,83,84]. However, WBV has some adverse effects, including dizziness, headaches, and falls. These can be minimized when exposure is of low intensity and for a short duration. Thus, it is important to select the vibration model and the duration of the intervention [53,85].

Although high-intensity, long-term RT is effective for improving muscle strength and mass [86,87], its effects on AS are conflicting because they increased carotid–femoral PWV in only one study out of three (Figure 3). On the other hand, if we start with low-intensity and continue to high-intensity, long-term RT caused no change in femoral-ankle or aortic PWV in middle-aged women [88,89] and young adults [44,90]. In healthy, active young males who had been doing high-intensity RT for at least the past two years, central AS declined. An important role in relation to varied arterial responses to RT, depends on age, health status, and training intensity [45]. The mechanism underlying increased AS with high-intensity RT remains unclear, but higher AS in young strength-trained men is associated with higher concentrations of plasma endothelin, a potent vasoconstrictor peptide produced by vascular endothelial cells. Another possibility is that the vigorous and frequent elevation of blood pressure during RT might engender AS [91,92]. According to Miyachi et al. [93], it is expected that values return to the baseline values within four months after the end of a long-term RT.

Aerobic exercise combined with RT brings further benefits for cardio-respiratory functions [94,95] and is typical in clinical practice, but currently it has no systematic evidentiary basis. One of the few studies in this regard is that of Guimaraes et al. [55], which found that a continuous and interval RT can help control blood pressure, but only interval training reduced arterial stiffness in hypertensive subjects undergoing treatment. Another interesting study was that by Figueroa et al. [61], in which 12-week moderate-intensity AE + RT training reduced arterial stiffness and improved hemodynamics, and muscle strength in previously sedentary postmenopausal women. Ashor et al. [16] concluded that AE improved AS significantly and that the effect was enhanced with higher AE intensity and in participants with greater AS at baseline. However, the previous reviews did not consider the exercise intensity and the location of PWV measurements as factors, which highly influenced the study results. 

This study was limited to the healthy population, but similar finding had been reported for postmenopausal woman [58] and patients with first-degree hypertension [80], metabolic syndrome [78] and cancer [79]. Some original studies included only a general description of the individual RT parameters. For this reason, it was difficult to clearly define optimal exercise selection such as bench press vs dumbbell press or leg press vs squat, or exercises order with rest interval. Despite all this, we managed to obtain a few basic parameters that played a key role (Table 3) in decreasing AS. However, it was necessary to verify specific training programs with a detailed description of the individual parameters, not only in general terms. Future studies should find out to what extent RT parameters, stretching and aerobics affected AS. Recent meta-analyses demonstrated that stretching reduced AS, HR, and DBP and improved endothelial function, all of which are crucial parameters of arteriosclerosis in middle-aged and older adults [96]. Another important thing that cannot be neglected is the location of PWV measurement. The carotid–femoral PWV measurement is the gold standard for AS testing due to the ease of measurement, reliability and the amount of evidence that has shown PWV to be associated with cardiovascular disease [27]. Figure 2 and Figure 3 present data for carotid–femoral PWV measurement. These types of PWV measurements were included only in seven studies that provided us with a large effect size. However, optimal carotid–femoral PWV is in the range of 6–9 ms^−1^, which was found only in Beck et al. [50] and Croymans et al. [44] in healthy normotensive adults after a resistance training intervention. Brachial-ankle PWV is a predictor of longitudinal increases in BP, cardiovascular events, cardiovascular disease and mortality [82,97]. This parameter positively affected AS in five RT studies, where only one showed no decrease (Figure 4).

This approach is one of the first distinguishing effects of different training intensities on PWV and its measuring methods. Intensity is essential for setting RT parameters such as the number of reps and sets, rest intervals, and exercise order and selection. The PWV should ensure standard values for evaluations.

### Practical Implications

A practical implication of this review is that the low and moderate intensity 30–80% 1RM should be used to reduce PWV and prevent cardiovascular diseases in healthy individuals during a long-term RT. We can recommend low intensity RT for beginners or individuals who are returning after a long time to RT, especially for safety reasons. On the other hand, moderate intensity RT is a great for individuals with RT experience, because they have a good knowledge of exercise technique. They can benefit from other benefits of RT such as increasing muscle mass and strength, improving bone density and body composition etc. [21,22]. Based on detailed analyses of original studies, such a low intensity RT should include 5–8 full body or split training that combine upper- and/or lower-body exercises (large and small muscle groups) at more than 10 repetitions per 3–5 sets. Frequency should be 2–4 per week for 6–12 weeks (Table 3). The vibration on WBV device can be horizontal at a frequency of 30 Hz (1 Hz = 1 oscillation/second), and the magnitude (acceleration) can be 3.2 *g* (gravity, 1 *g* = 9.81 m/s 2) in healthy adults. We can include bodyweight exercises (squat, lunges, push-ups, etc.) and dumbbell exercises (biceps curls, triceps extensions, shoulder presses, etc.) with low intensity RT parameters.

The main difference between low and moderate intensity RT is in the number of repetitions, the rest period and exercise order (Table 3). Another difference is in exercise order, where upper body exercises at moderate intensity can increase AS, therefore, upper body exercise should be combined with lower body, full body exercises or other non-consecutive order. All suggested RT parameters that contributed to the decrease in AS, can be easily applied by movement specialists, fitness trainers and physiotherapists in a healthy adult population, regardless of sex.

## 5. Conclusions

High-intensity RT (above 80% 1RM) with maximal and submaximal resistance might have positive or negative effect on AS, which means that current research should focus on setting other loading parameters that can decrease PWV when using high intensity RT. Moderate intensity (60–80% 1RM), the most common intensity, decreased carotid–femoral PWV. On the other hand, future research should find out how to avoid increased brachial–ankle PWV during an upper limb RT. Low-intensity RT or WBV training has a positive effect on AS and other cardiorespiratory functions regardless of the method of PWV measurement. WBV training, which uses body weight or free weights during an RT bout, decreased AS in all PWV measurements (carotid–femoral PWV, femoral–ankle PWV, brachial-ankle), lowered blood pressure and improved body composition similarly to a low-intensity RT. In addition, we included AE alone as part of a long-term RT because it is very well researched and recommended for the secondary prevention of cardiovascular disease, especially in patients with high blood pressure and AS.

## Figures and Tables

**Figure 1 jcm-10-03492-f001:**
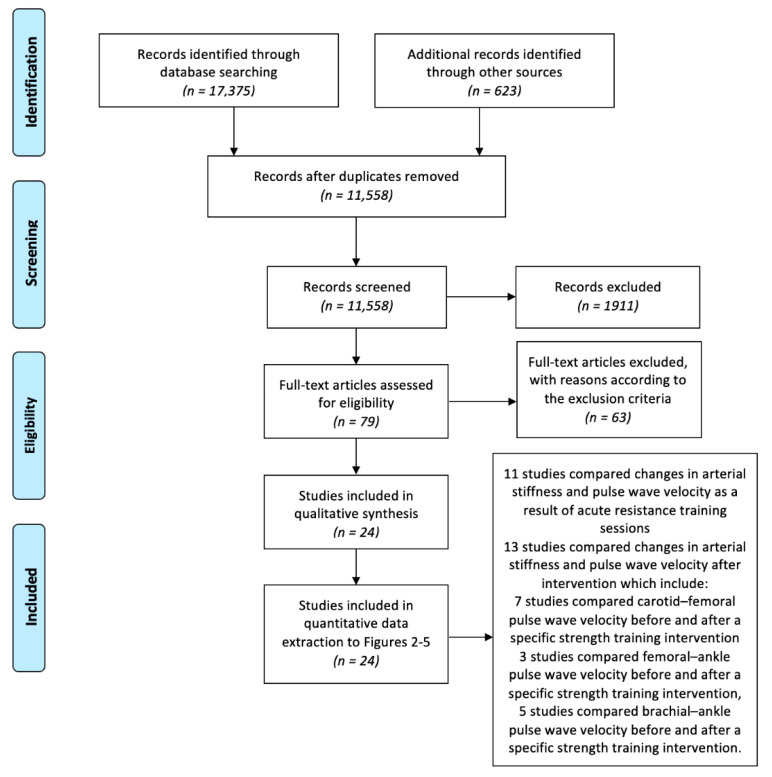
Flowchart of the selection process.

**Figure 2 jcm-10-03492-f002:**
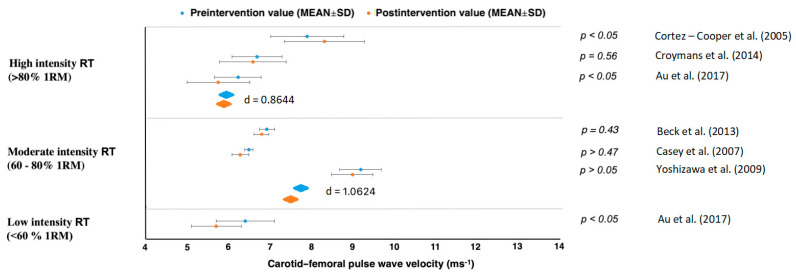
Carotid–femoral pulse wave velocity changes.

**Figure 3 jcm-10-03492-f003:**
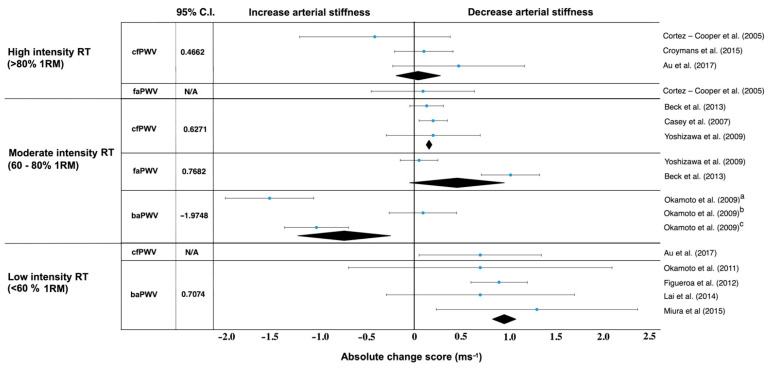
Absolute change score (ms^−1^) in long-term resistance training programs. ^a^ upper limb training group, ^b^ lower limb training group, ^c^ group of resistance training with slow lifting and quick lowering. baPWV, brachial–ankle pulse wave velocity; caPWV, carotid–femoral pulse wave velocity; faPWV, femoral–ankle pulse wave velocity; RM, repetition maximum. Black diamonds represent the weighted mean and standard error, blue dot is the mean difference with standard deviation.

**Figure 4 jcm-10-03492-f004:**
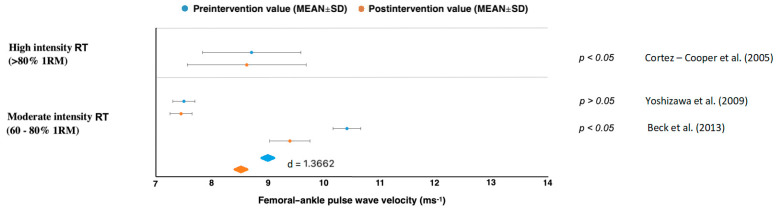
Femoral–ankle pulse wave velocity changes. RM = repetition maximum, SD = standard deviation.

**Figure 5 jcm-10-03492-f005:**
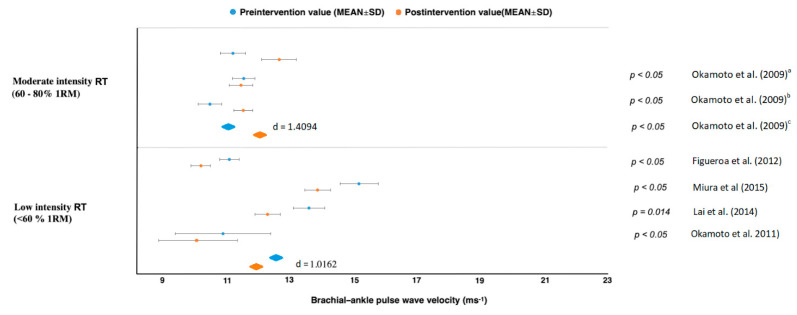
Brachial–ankle pulse wave velocity changes. ^a^ upper limb training group, ^b^ lower limb training group, ^c^ group of resistance training with slow lifting and quick lowering.

**Table 1 jcm-10-03492-t001:** Changes in arterial stiffness and pulse wave velocity as a result of acute resistance training bout.

Authors	Subjects	PWV Measurement	Acute Resistance Training Bout and Duration	Results
**High Intensity RT Bout (>80% 1RM)**				
Barnes et al. (2010) [34]	*N* = 27 men, healthy, young	cfPWV	Eccentric RT bout: 1. Leg press (bilateral): 6 sets of 10 repetitions at 110% 2. Elbow flexion (unilateral): 2 sets of 20 unilateral eccentric elbow flexion contractions	48 h after leg RT (*p* < 0.05) and arm RT ↑cfPWV (*p* < 0.05).
Lefferts et al. (2015) [40]	*N* = 20, healthy, recreationally active	cfPWV	5 sets, 5 repetition of maximum bench press; 5 sets of 10 repetition maximum biceps curls	An acute high intensity RT had ↑AS and ↑extracranial pressure pulsatility.
**Moderate Intensity RT_Bout (60–80% 1RM)**				
DeVan et al. (2005) [33]	*N* = 16, mix, healthy, young	Beta stiffness index	9 full body exercises, 75% 1RM	RT bout immediately and 30 min after had ↓carotid arterial compliance and ↑stiffness index. These values returned to baseline by 60 min.
Yoon et al. (2010) [35]	*N* = 13 healthy men	cfPWV	8 full body exercises, 60% 1RM	An acute RT bout had ↑AS. These values returned to baseline by 20 min.
Nitzsche et al. (2016) [39]	*N* = 41, mix healthy, physically active	cfPWV	RT bout: Group 70% 1RM	An acute moderate RT bout had ↓AS and ↓central and systolic BP.
Kingsley et al. (2017) [38]	*N* = 13, mix, healthy, recreationally resistance trained	cfPWV	Free-weight RT: 75% 1RM, 3 sets, 10 repetitions	An acute RT bout had ↑augmentation index and ↑AS without significantly altering aortic BP.
Tomschi et al. (2018) [36]	*N* = 20 healthy women	cfPWV	A. Upper body B. Lower body 12 repetitions with 70% 1RM	The adaptation pattern of the measured PWV as a measure of AS parameters in upper body RT bout compared to lower body RT bout is similar, and all parameters regulate to their baseline values within 60 min.
Parks et al. (2020) [42]	*N* = 32, young, healthy individuals	cfPWV	Free-weight RT bout VS weight machines RT bout: 3 sets of 10 repetitions at 75% 1RM	An acute free-weight and weight-machine RT bout are associated with transient ↑pulse wave reflection and ↑AS.
Rodríguez-Perez et al. (2020) [41]	*N* = 32, physically active, normotensive, and experienced with RT	cfPWV	3 sets at 75% 1RM, bench press and squat Group 1: high-effort Group 2: low-effort	Both training groups immediately after acute RT bout reported ↑AS and ↑BP while BP and AS returned to baseline levels 5 min and 24 h after completing the RT bout.
**Low Intensity RT Bout (<60% 1RM)**				
Okamoto et al. (2014) [37]	*N* = 10, mix, healthy	Beta stiffness index	bench press, 40% of 1RM, 3 sets	Carotid arterial compliance and the β-stiffness index significantly ↑ and ↓, respectively (both *p* < 0.05), at 30 and 60 min after the acute low intensity RT bout.
Nitzsche et al. (2016) [39]	*N* = 41, mix healthy, physically active	cfPWV	Squats, bench press, rowing with the barbell, biceps curl with the SZ curl bar, lying triceps extensions with the SZ curl bar 1. Group—3 sets at 30% 1RM, 30 repetitions 2. Group—3 sets at 50% 1RM, 20 repetitions 3. Group—4 sets, 70% 1RM, 10 repetitions	An acute moderate RT bout had ↑AS and ↓central systolic BP.

AS arterial stiffness, BP blood pressure, cfPWV carotid-femoral pulse wave velocity, PWV pulse wave velocity, RM repetition maximum, RT resistance training.

**Table 2 jcm-10-03492-t002:** Changes in arterial stiffness and pulse wave velocity as a result of a long-term resistance training program.

Authors	Subjects	PWV Measurement	Frequency, Load and Duration of RT	Results
**High-Intensity RT (** **>** **80% 1RM)**				
Cortez-Cooper et al. [43]	n = 23 healthy women, young	cfPWV, faPWV	11 weeks, 4× per week	Long-term high-intensity RT had ↑AS and ↑wave reflection.
Croymans et al. [44]	n = 36, overweight and obese men	cfPWV	12 weeks, 3× per week	Long-term high-intensity RT had no effect on augmentation index (*p* = 0.34) and cfPWV (*p* = 0.43).
Au et al. [45]	n = 16, healthy, active males	cfPWV	(heavier-load, lower-repetition), 12 weeks, 4× per week	↓Central arterial stiffness after RT, regardless of the load lifted.
**Moderate-Intensity RT (** **60–** **80% 1RM)**				
Casey et al. [46]	n = 30, healthy young adults	cfPWV	Progressive RT: 12 weeks, 3× per week	RT consisting of progressively higher intensity without concurrent increases in training volume does not increase central or peripheral AS or alter aortic pressure wave characteristics.
Okamoto et al. [47]	n = 30, mix, healthy, young	baPWV	Upper and lower limb RT: 10 weeks, 2× per week	upper limb RT had ↑baPWV (*p* < 0.05). In contrast, baPWV in the lower limb RT had not changed from baseline.
Okamoto et al. (2009) [48]	n = 30, healthy men	baPWV	10 weeks, 2× per week	RT with prolonged eccentric phase did not change from baseline baPWV. In contrast, RT with prolonged concentric phase had ↑baPWV.
Yoshizawa et al. [49]	n = 11 healthy, sedentary middle-aged women	cfPWV, faPWV	12 weeks, 2× per week	Long-term moderate-intensity RT did not increase AS.
Beck et al. [50]	n = 43 mix, prehypertension, without medication	cfPWV, faPWV	8 weeks, 3× per week	Long-term RT and AE alone effectively had ↓peripheral AS, ↓augmentation index.
**Low-Intensity RT (** **<60% 1RM)**				
Okamoto et al. [51]	n = 13, mix, young healthy adults	baPWV	Low-intensity RT with short inter-set rest period: 10 weeks, 2× per week	Long-term low-intensity RT with short inter-set rest period had ↓AS and ↓baPWV.
Figueroa et al. [52]	n = 10 young, overweight or obesity, women	baPWV	Whole-body vibration RT: 6 weeks, 3× per week	Long-term WBV-RT had ↓systemic AS via improvements in wave reflection and sympathovagal balance.
Lai et al. [53]	n = 38, mix, adult and older adults	baPWV	Whole-body vibration training, 3 months, 3× per week	Long-term WBV-RT had ↓AS.
Miura et al. [54]	n = 100 women, normotension	baPWV	circuit RT, 12 weeks, 2× per week	Long-term RT had fewer↓ASs.
Au et al. [45]	n = 16, healthy, active males	cfPWV	lighter-load, higher-repetition, 12 weeks, 4× per week	Long-term RT had ↓central AS, regardless of the load lifted.

AE, aerobic exercise; PWV, pulse wave velocity; baPWV, branchial–ankle PWV; cfPWV, carotid–femoral PWV; faPWV, femoral–ankle PWV; RT, resistance training; SBP, systolic blood pressure; WBV, whole-body vibration training.

**Table 3 jcm-10-03492-t003:** Long-term resistance training recommendations for arterial stiffness decrease in healthy individuals.

Type of RE	Intensity	Sets	Reps	Rest	Duration	Frequency	Exercises Selection	Exercises
Low intensity RE	30–60% 1RM	3–5	>10 and higher repetition)	30–60 s	6–12 week	2–4× per week	5–8 exercises, weight-machine RE, free-weight RE, circuit RE, WBV RE	chest press, shoulder press, biceps curl, triceps extensions, seated row, lateral pull down, leg press, leg extension, leg curl, front-plank and sit-ups
Moderate intensity RE	60–80% 1RM	2–5	8–12	90–180 s	8–12 week	2–3× per week	5–7 exercises, weight-machine RE, free-weight RE, upper body exercises not in consecutive order.	chest press, shoulder press, biceps curl, seated row, lateral pull down, squat, leg press, leg extension, leg curl, hip adduction, calf raises and sit-ups

RT resistance training, RM repetition maximum, WBV whole-body vibration training.

## Data Availability

The data are presented in Appendix A.

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
