# Peer review of "Effect of an Acute Resistance Training Bout and Long-Term Resistance Training Program on Arterial Stiffness: A Systematic Review and Meta-Analysis"

_jcm, 2021, doi:10.3390/jcm10163492_

Round 1
Reviewer 1 Report
The systematic review with meta-analysis presented is very well done and very complete.
I would like to make some small recommendations to the authors:
- On page 3, place PEDro (Physiotherapy Evidence Database) scale and its reference (Moseley, Herbert, Sherrington and Maher, 2002).
- You should include a table with the methodological quality criteria and explain them By PEDro score
- Please check in the discussion (page 11) “However, these results remain to be confirmed in other age groups, females, or cardiovascular patients.[41]” and (Page 12) “Another possibility is that vigorous and frequent elevation of blood pressure during resistance exercise might engender arterial stiffening.[88,89]”and (Page 13) “In the future, it will be necessary to find out to which extent the strength training parameters, stretching and the aerobic part of training have an effect on arterial stiffness. Recent meta-analyses demonstrated that stretching exercises reduced arterial stiffness, HR, and DBP, and improved endothelial function, which are crucial parameters of arteriosclerosis in middle-aged and older adults. [99] “, references must be placed in front of the punctuation mark
- In different pages there are different font sizes
- It is recommended that they include a section on practical implications.
- In addition, you should write down the strengths and limitations.
Author Response
The systematic review with meta-analysis presented is very well done and very complete.
Answer: Thank you for your positive evaluation. We have done all changes which you suggested.
I would like to make some small recommendations to the authors:
On page 3, place PEDro (Physiotherapy Evidence Database) scale and its reference (Moseley, Herbert, Sherrington and Maher, 2002).
Answer: We have explained the abbreviation and suggested reference has been added.
You should include a table with the methodological quality criteria and explain them By PEDro score.
Answer: We added the link to the supplementary file with whole PEDro score table and the items explanation below. We have decide to not disrupt the main text by this table.
Please check in the discussion (page 11) “However, these results remain to be confirmed in other age groups, females, or cardiovascular patients.[41]” and (Page 12) “Another possibility is that vigorous and frequent elevation of blood pressure during resistance exercise might engender arterial stiffening.[88,89]”and (Page 13) “In the future, it will be necessary to find out to which extent the strength training parameters, stretching and the aerobic part of training have an effect on arterial stiffness. Recent meta-analyses demonstrated that stretching exercises reduced arterial stiffness, HR, and DBP, and improved endothelial function, which are crucial parameters of arteriosclerosis in middle-aged and older adults. [99] “, references must be placed in front of the punctuation mark
Answer: Done, we have doublecheck the dot positions.
In different pages there are different font sizes
Answer: Thank you, we have revised the font size.
It is recommended that they include a section on practical implications.
Answer: The practical implications has been added as subchapter of discussion.
In addition, you should write down the strengths and limitations.
Answer: Strength and limitation has been added to the last paragraph of discussion.
Reviewer 2 Report
This systematic review and meta-analysis aimed to how acute resistance exercise and long-term resistance exercise programs impact arterial stiffness, as measured by pulse wave velocity (PWV). There was also an investigation into the impact of exercise intensity on this outcome. Although this paper has the potential to contribute to the field, I have outlined a number of changes that need to be made prior to publication.
Main comments:
- Strongly recommend copy editing throughout for sentence structure and word selection
- Overall, I found the review hard to follow in terms of the overall aims and the results. A couple of suggestions of how to improve this
- Use of the same terms to describe an acute resistance exercise bout versus a long- term resistance exercise program throughout the entire manuscript, including the I found myself going back and forth to figure out which type of training you were talking about.
- Please clarify your aims (mostly via improved sentence structure). As I understand it, the aims were two-fold: In healthy people, this review aimed to a) understand the changes in arterial stiffness with an acute resistance exercise bout, as well as a long- term resistance exercise program, and b) determine the impact of exercise intensity on these changes.
- Avoid discussion about populations other than your target population (i.e., healthy adults) – you continually bring up populations such as hypertension, peripheral arterial disease, menopause etc.; this is not helpful or applicable to your review. You can of course discuss the implications of reducing increases in arterial stiffness in terms of preventing cardiovascular disease, but resistance training (both acute and long-term) may result in different changes in these other populations than it does in healthy adults. I note in your abstract you say individuals with or without cardiovascular disease, but there is no separation of those in the article?
- There are times where you go way off topic e.g., in the discussion: “Therefore, it is not recommended for beginners and special populations because the techniques are difficult and notably increase the probability of injury”. You should revise your discussion and ensure you are including only what applies to the
- How does your review add to the literature? As you noted, there have been a number of systematic reviews/meta-analyses in the area and although you say you focus on exercise intensity, so did Miyachi et al 2013 and Ashor et al 2014… if you decide that the method of PWV measurement is your distinguishing factor, this needs to be discussed more throughout the manuscript. I acknowledge that these reviews are somewhat dated, but how many papers are you adding to the analysis? You could discuss this. You also mention the combined training should be analysed, which I agree with, but as noted below, there were only two studies you included, so you haven’t really done this. Perhaps looking at changes based on baseline PWV and/or dose of exercise prescribed? Although using different outcomes, see the really interesting analysis done by Patten et a. 2020 Front Physiol 2020; 11:606 “Exercise Interventions in Polycystic Ovary Syndrome: A Systematic Review and
Meta-Analysis”. Or you could add central haemodynamics (measured by pulse wave analysis)? You don’t need to/shouldn’t do all of these, but just some suggestions.
Abstract
- Says you are comparing the effect of acute and “chronic” training on arterial stiffness, but no mention of intensity comparison? And you aren’t really comparing them, but rather understanding how each time effects PWV.
Introduction
- You talk about the resistance training guidelines for people with peripheral arterial disease, but you are reviewing the effects in healthy people, so you should focus on those guidelines
- Overall, I think this section could be cut down significantly – bring in a few key messages to rationalise your study (e.g., resistance training recommended for healthy adults for general health, including cardiovascular health à PWV indicative of cardiovascular health à unknown how PWV changes with an acute vs. resistance program, and the impact of exercise intensity)
- In the final two sentences, you say “places” – would say methods
Methods
- You said you included “all adult populations” in the review – is that correct? If so, as above, I am unsure that is
Results
- The fact you were able to identify an additional 600+ records from other sources is a red flag
– your search terms need to be optimised until you capture as many of those as possible
- In tables 1 & 2, I would recommend reducing the words in the results column and simplifying it so arrows up and down for changes in PWV only. Since you also wanted the method of PWV measurement to be a distinguishing factor of this review, it would be good to include those details in the table as
- I would recommend redoing the Forrest plots in figures 2-4 so that you present the absolute change score, rather than the pre-post values. It will make it a lot easier to understand
Discussion
- I would like to see more of a discussion about the mechanisms behind the different intensities impacting PWV in different ways, as well as why looking at acute vs. exercise programs is important
- Do you have a reference to say that the changes in arterial stiffness with acute exercise are likely functional not structural?
Supplementary Material
- Given only two included studies investigated combined aerobic and resistance training (Guimaraes et al. (2010); Figueroa et al. (2011)) and the point of the review was to investigate the effects of resistance training alone, it would be better to exclude those studies from the analysis so as not to confuse the message

Author Response
Strongly recommend copy editing throughout for sentence structure and word selection
Answer: We have send this article to MDPI English editing service, so this editing has been done.
Overall, I found the review hard to follow in terms of the overall aims and the results. A couple of suggestions of how to improve this
Answer: Thank you for you detailed review and helpful recommendation. We have revised our manuscript according to your suggestions.
- Use of the same terms to describe an acute resistance exercise bout versus a long- term resistance exercise program throughout the entire manuscript, including the I found myself going back and forth to figure out which type of training you were talking about.
Answer: We have clarified the use of terminology throughout the manuscript.
- Please clarify your aims (mostly via improved sentence structure). As I understand it, the aims were two-fold: In healthy people, this review aimed to a) understand the changes in arterial stiffness with an acute resistance exercise bout, as well as a long- term resistance exercise program, and b) determine the impact of exercise intensity on these changes.
Answer: We have revised the aim according to your suggestion, highlighting two major scopes and aim to consider measurement at different body location and focus only for healthy people.
- Avoid discussion about populations other than your target population (i.e., healthy adults) – you continually bring up populations such as hypertension, peripheral arterial disease, menopause etc.; this is not helpful or applicable to your review. You can of course discuss the implications of reducing increases in arterial stiffness in terms of preventing cardiovascular disease, but resistance training (both acute and long-term) may result in different changes in these other populations than it does in healthy adults. I note in your abstract you say individuals with or without cardiovascular disease, but there is no separation of those in the article?
Answer: Upon your comments, we have decided to remake whole review only for healthy population, which clear out the manuscript. We are still mentioning studies on other population in discussion and article selection. However, the main conclusions are strictly for the healthy individuals.
Thus, we have excluded from the article (Results, Tables, Figures) population with hypertension, peripheral arterial disease, menopause etc. In Discussion we discussed the implications of reducing increases in arterial stiffness in terms of preventing cardiovascular disease in population with hypertension, peripheral arterial disease and menopause.
- There are times where you go way off topic e.g., in the discussion: “Therefore, it is not recommended for beginners and special populations because the techniques are difficult and notably increase the probability of injury”. You should revise your discussion and ensure you are including only what applies to the
Answer: We have changed the discussion a lot. Focusing much less on diseases, and focusing more on clinical recommendations for healthy people.
How does your review add to the literature? As you noted, there have been a number of systematic reviews/meta-analyses in the area and although you say you focus on exercise intensity, so did Miyachi et al 2013 and Ashor et al 2014… if you decide that the method of PWV measurement is your distinguishing factor, this needs to be discussed more throughout the manuscript.
Answer: Thank you for this point. We highlighted in the introduction, that our distinguishing factor is the resistance exercises intensity together with PWV measurement.
We have point out to inconsistency of previous reviews on those issues, where previous reviews did not work with exercise intensity as main (key) separator, sometimes just approximate values, and did not consider PWV measure at all (although they reported kind of measure in description table).
We applied this approach in this manuscript.
I acknowledge that these reviews are somewhat dated, but how many papers are you adding to the analysis? You could discuss this.
Answer: The previous reviews are some dated and also do not rigorously work with the issue of exercise intensity, nor with method of PWV measurement (body part). We are adding 8 long-term RE program studies upon the Miyachi et al. 2013 study, 8 long-term RE program studies and 5 acute RE bout studies upon the García-Mateo et al. 2020 study, 7 long-term RE program studies upon the Ceciliato et al. 2020 study and 6 acute RE bout studies upon the Saz-Lara et al. (2021) study.
On the other hand we did not included some studies as previous reviews due to more strict selection approach.
You also mention the combined training should be analysed, which I agree with, but as noted below, there were only two studies you included, so you haven’t really done this. Perhaps looking at changes based on baseline PWV and/or dose of exercise prescribed?
Answer: We have revised the selected file to only healthy people and there is now only one paper which dominantly included resistance training with low amount of aerobic training. The combine training is now only mentioned in the discussion.
Although using different outcomes, see the really interesting analysis done by Patten et a. 2020 Front Physiol 2020; 11:606 “Exercise Interventions in Polycystic Ovary Syndrome: A Systematic Review and Meta-Analysis”. Or you could add central haemodynamics (measured by pulse wave analysis)? You don’t need to/shouldn’t do all of these, but just some suggestions.
Answer: We have look at this review and get some inspiration out of it. We used this reference also in discussion.
Abstract
- Says you are comparing the effect of acute and “chronic” training on arterial stiffness, but no mention of intensity comparison? And you aren’t really comparing them, but rather understanding how each time effects PWV.
Answer: We agree, thus we revised the aim according to your suggestion.
Introduction
- You talk about the resistance training guidelines for people with peripheral arterial disease, but you are reviewing the effects in healthy people, so you should focus on those guidelines.
Answer: We agree, therefore we focus primarily on healthy people with separation of intensity is now referring both healthy and aging population.
- Overall, I think this section could be cut down significantly – bring in a few key messages to rationalise your study (e.g., resistance training recommended for healthy adults for general health, including cardiovascular health à PWV indicative of cardiovascular health à unknown how PWV changes with an acute vs. resistance program, and the impact of exercise intensity)
Answer: We have cut down the introduction a lot, Now is much more straight forward.
- In the final two sentences, you say “places” – would say methods
Answer: Done
Methods
- You said you included “all adult populations” in the review – is that correct? If so, as above, I am unsure that is
Answer: Sorry for this typo. We corrected the statement to the adult healthy population.
Results
- The fact you were able to identify an additional 600+ records from other sources is a red flag
– your search terms need to be optimised until you capture as many of those as possible.
Answer: We have to point out that one of the “additional source” was the Google Scholar, which is the main cause of this relatively high number. Since Google Scholar have machine learning strategy to search, we cannot put this approach to standardized database source with full replicable search.
Upon many discussion with librarian, we have decided to leave our search formula rather more wide to not miss any article. More detailed specification and optimization of formula would result in some article loss. Because we used this strategy we have ended up with 11,530 records for manual title and abstract screening. Which is higher that typical standard, but we believe that for such multidisciplinary topic is better to cover all possibilities. Also, the Google Scholar search was quite challenge to screen, but we believe that it strengthened the search strategy.
- In tables 1 & 2, I would recommend reducing the words in the results column and simplifying it so arrows up and down for changes in PWV only. Since you also wanted the method of PWV measurement to be a distinguishing factor of this review, it would be good to include those details in the table as
Answer: We agree, thus we have shortened both tables. And we have add the PWV method to them.
- I would recommend redoing the Forrest plots in figures 2-4 so that you present the absolute change score, rather than the pre-post values. It will make it a lot easier to understand.
Answer: According to your suggestion we have added figure 5 showing absolute change score and weighted means. Now there are all results in one final graph. However we have left pre-post values graphs because they includes d effect counted from difference in control groups.
Discussion
- I would like to see more of a discussion about the mechanisms behind the different intensities impacting PWV in different ways, as well as why looking at acute vs. exercise programs is important.
Answer: We agree highlight those issues in the discussion.
- Do you have a reference to say that the changes in arterial stiffness with acute exercise are likely functional not structural?
Answer: We added a reference - Saz-Lara et al. (2021)[74].
Supplementary Material
- Given only two included studies investigated combined aerobic and resistance training (Guimaraes et al. (2010); Figueroa et al. (2011)) and the point of the review was to investigate the effects of resistance training alone, it would be better to exclude those studies from the analysis so as not to confuse the message.
Answer: We agree that our reference list do not include enough combine RE and other training methods. We have excluded those studies on the criteria of healthy people.
Round 2
Reviewer 2 Report
Thank-you for your thorough response and changes to the document. Overall, these changes have significantly improved the manuscript, so well-done. A few final notes:
- I would recommend another once over by a copy editor - there are still a number of grammatical errors/sentence structure that could be optimised to improve readability
- I would make the title "...exercise bout AND long-term..."
- I prefer the acronym RT (resistance training) over RE. This is up to you though.
- In the abstract, you have the following sentence: "A long-term RE program at above an 80% repetition maximum (RM) increased carotid–femoral PWV (Cohen’s d = 0.86) by about 0.42 ± 0.8 ms-1 (p < 0.05) but also decreased it up to 0.47 ± 0.7 ms-1 (p < 0.05)." How can it increase and decrease PWV?
- There's a caption for figure 5 on page 6/7, but no figure... I can see the figure is on page 9 and already has a caption. This new figure is excellent, well done!
- Why do you think baPWV was increased with moderate-intensity exercise versus the other methods? Would be good to discuss this given it is part of your study aims
- At the end of the first paragraph in the discussion, you mention the "body part" being important but it's just a side mention with no discussion. And it interrupts the discussion about the long-term vs acute exercise. I would recommend re-ordering slightly to help with flow, and do so in line with what you discuss in the discussion (i.e., summarise the acute effects first, then make that the first paragraph of the discussion, and so on)
- I really like the inclusion of the Practical Implications section. However, it doesn't match the conclusions section...For example, if moderate intensity is beneficial for PWV, then why would people prioritise low intensity (which we know may not be as beneficial for other health outcomes e.g. fitness)?
- I am unsure of the benefit of table 3 as you provide all of the information in text
Author Response
Thank-you for your thorough response and changes to the document. Overall, these changes have significantly improved the manuscript, so well-done.
Answer: Thank you for your positive evaluation. We make further improvement of our manuscript.
A few final notes:
- I would recommend another once over by a copy editor - there are still a number of grammatical errors/sentence structure that could be optimized to improve readability
Answer: The English will by edited once again during editorial process.
- I would make the title "...exercise bout AND long-term..."
Answer: We agree, thus we change the title as suggested.
- I prefer the acronym RT (resistance training) over RE. This is up to you though.
Answer: We finally agree and revised whole manuscript to use abbreviation RT – resistance training.
- In the abstract, you have the following sentence: "A long-term RE program at above an 80% repetition maximum (RM) increased carotid–femoral PWV (Cohen’s d = 0.86) by about 0.42 ± 0.8 ms-1 (p < 0.05) but also decreased it up to 0.47 ± 0.7 ms-1 (p < 0.05)." How can it increase and decrease PWV?
Answer: True is that according to our result the high intensity training have an ambiguous effect on PWV, which is now stated in abstract instead reporting contradictory values.
- There's a caption for figure 5 on page 6/7, but no figure... I can see the figure is on page 9 and already has a caption. This new figure is excellent, well done!
Answer: We are glad for your previous suggestion to cover the results by such figure. This caption was left there accidentally since a lot of track changes has been made. Now we simply deleted this redundant caption.
- Why do you think baPWV was increased with moderate-intensity exercise versus the other methods? Would be good to discuss this given it is part of your study aims.
Answer: Thang you for this point, we have added this issue into the discussion.
- At the end of the first paragraph in the discussion, you mention the "body part" being important but it's just a side mention with no discussion. And it interrupts the discussion about the long-term vs acute exercise. I would recommend re-ordering slightly to help with flow, and do so in line with what you discuss in the discussion (i.e., summarise the acute effects first, then make that the first paragraph of the discussion, and so on)
Answer: We have re-order the beginning of body part discussion in the text and added more to this issue toward practical application.
- I really like the inclusion of the Practical Implications section. However, it doesn't match the conclusions section...For example, if moderate intensity is beneficial for PWV, then why would people prioritize low intensity (which we know may not be as beneficial for other health outcomes e.g. fitness)?
Answer: We have now explained more the approach and context of the practical implication. Especially we explained what should be the exercise intensity progress.
- I am unsure of the benefit of table 3 as you provide all of the information in text
Answer: We understand your comment and now he left the summarized guidelines in the text, but we put in table 3 separate guidelines for moderate and low intensity RT. Moreover, we more introduces the usefulness of this table in this part.
We believe that this table has a important meaning since this form of training prescription is quite standardized and easy to understand for movement therapists, physiotherapists and other coaches.